# What Are the Key Factors for the Detection of Peptides Using Mass Spectrometry on Boron-Doped Diamond Surfaces?

**DOI:** 10.3390/nano14151241

**Published:** 2024-07-24

**Authors:** Juvissan Aguedo, Marian Vojs, Martin Vrška, Marek Nemcovic, Zuzana Pakanova, Katerina Aubrechtova Dragounova, Oleksandr Romanyuk, Alexander Kromka, Marian Varga, Michal Hatala, Marian Marton, Jan Tkac

**Affiliations:** 1Institute of Chemistry, Slovak Academy of Sciences, 841 04 Bratislava, Slovakia; 2Institute of Electronics and Photonics, Faculty of Electrical Engineering and Information Technology, Slovak University of Technology, 841 04 Bratislava, Slovakia; marian.vojs@gmail.com (M.V.);; 3Centre of Excellence for Glycomic, Slovak Academy of Sciences, 841 04 Bratislava, Slovakia; 4FZU—Institute of Physics, Czech Academy of Sciences, Cukrovarnická 10, 162 00 Prague, Czech Republickromka@fzu.cz (A.K.); 5Institute of Electrical Engineering, Slovak Academy of Sciences, 841 04 Bratislava, Slovakia; 6Department of Graphic Arts Technology and Applied Photochemistry, Faculty of Chemical and Food Technology, Slovak University of Technology, 812 37 Bratislava, Slovakia

**Keywords:** boron-doped diamond, surface termination, peptides, MALDI, mass spectrometry, nanostructures

## Abstract

We investigated the use of boron-doped diamond (BDD) with different surface morphologies for the enhanced detection of nine different peptides by matrix-assisted laser desorption/ionisation mass spectrometry (MALDI-MS). For the first time, we compared three different nanostructured BDD film morphologies (Continuous, Nanograss, and Nanotips) with differently terminated surfaces (-H, -O, and -F) to commercially available Ground Steel plates. All these surfaces were evaluated for their effectiveness in detecting the nine different peptides by MALDI-MS. Our results demonstrated that certain nanostructured BDD surfaces exhibited superior performance for the detection of especially hydrophobic peptides (e.g., bradykinin 1–7, substance P, and the renin substrate), with a limit of detection of down to 2.3 pM. Further investigation showed that hydrophobic peptides (e.g., bradykinin 1–7, substance P, and the renin substrate) were effectively detected on hydrogen-terminated BDD surfaces. On the other hand, the highly acidic negatively charged peptide adrenocorticotropic hormone fragment 18–39 was effectively identified on oxygen-/fluorine-terminated BDD surfaces. Furthermore, BDD surfaces reduced sodium adduct contamination significantly.

## 1. Introduction

Matrix-assisted laser desorption/ionisation mass spectrometry (MALDI-MS) is an effective tool for the analysis of (bio)molecules of interest in the desorption/ionisation process to produce positively charged single ions [1]. MALDI-MS is extensively used for the analysis of medically relevant biomolecules, such as proteins, post-translational modifications of proteins, and peptides. Its high-throughput capabilities, facilitated by soft ionisation, which minimises fragmentation and enables the production of singly charged ions, are particularly advantageous. Moreover, MALDI-MS exhibits robust tolerance for contaminants, such as salts, buffers, and other substances, commonly present in biological samples, thereby making it a preferred method over ESI ionisation [2,3]. Peptides have important functions in cellular communication, and they can be used as candidates for biomarker discovery and precise disease diagnosis [4]. The peptides selected for this study are commonly used as calibration standards for MS analysis. Their defined characteristics, representing a wide range of molecular weights, hydrophobicity, and pI values, ensure reproducibility and enable the comparison of different methods across various laboratories. Currently, one of the most studied techniques for small biomolecule detection is based on the application of nanostructured surfaces [5]. Recently novel nanostructured steel plates, possessing large surfaces, laser absorption efficiencies, and high ionisation efficiencies, have been used for the analysis of small and large molecules [1].

A porous silicon layer was successfully used as a support, resulting in detection levels of the analyte as low as femtomoles (or attomoles), with little or no analyte fragmentation [6]. Similarly, a highly porous activated carbon layer deposited on an alumina support was shown as a feasible alternative to silicon layers [7]. The recorded mass spectra were almost clean, with only a few background ion peaks appearing at low masses. Both these results and the rapid progress in nanomaterial synthesis have prompted researchers to test a variety of nanomaterials, such as metallic nanoparticles (Au and Ag), metal oxides (ZnO and Fe_3_O_4_), metal-Organic frameworks, and sp^2^-hybridised carbon nanoforms [8,9,10,11,12,13,14,15].

Diamond, sp^3^-hybridised carbon, possesses an extraordinary combination of intrinsic properties, including high hardness, excellent thermal conductivity, chemical inertness to harsh conditions, and tuneable surface chemistry (i.e., the ability to tailor the surface charge and electron affinity) [16]. Moreover, its electrical properties can be tuned from insulating to semi-metallic through doping with boron [17,18]. The surface morphology of diamond can be technologically controlled, ranging from micro- to nanoscale structural features. Deep nanoporous etches can be easily made by reactive ion etching [19,20] or a novel process called molten salt thermal etching [21]. Additionally, its surface wettability can be varied through atomic termination with elements, such as oxygen, hydrogen, and fluorine, or amine groups [22]. In this view, diamond appears as a promising platform for MALDI-MS, which has not yet been explored. 

In this article, we provide an application of the BDD interface in bioanalysis because BDD can be deposited with excellent uniformity [23]. BDD was applied for MALDI-MS analysis, with a focus on assays of peptides for the first time.

## 2. Experimental Section

### 2.1. Chemicals

The MALDI matrix (2,5-dihydroxybenzoic acid (DHB)) was purchased from Sigma-Aldrich (Taufkirchen, Germany) and trifluoroacetic acid (TFA) from Fisher Scientific (Loughborough, UK) and acetonitrile (ACN) from Merck KGaA (Taufkirchen, Germany) were purchased from Honeywell (Offenbach am Main Germany). The nine peptides used as standards for MS were purchased from Sigma-Aldrich (Taufkirchen, Germany). Peptides bradykinin 1–7 (Brad), angiotensin II (Ang II), angiotensin I (Ang I), bombesin (Bomb), renin substrate (Ren S), adrenocorticotropic hormone fragment 1–17 (ACTH 1–17; AC1), adrenocorticotropic hormone fragment 18–39 (ACTH 18–39; AC18), and somatostatin (Som) were provided from Sigma-Aldrich (Taufkirchen, Germany), and substance P (Sub P) from Hello Bio (Bristol, UK). The peptides were selected based on the preliminary experiment with Bruker’s peptide standard (consisting of all the peptides described above).

### 2.2. Diamond Growth and Structuring

Boron-doped diamond (BDD) films were grown on polished Si (100) and Al_2_O_3_ substrates; cleaned in acetone, isopropyl alcohol, and DI (deionised) water; and ultrasonically seeded with diamond nanoparticles (5 nm) in a DI water suspension (50 mg L^−1^). Depositions were carried out in a linear antenna microwave (MW) plasma chemical vapour deposition (CVD) reactor (Cube 300, scia Ltd., Chemnitz, Germany), using 6 kW of microwave power per pulse (2 × 3 kW, with power on and off set at 8 and 6 ms, respectively, and a 50% phase change for each of the two antennas), for 30 h at a substrate temperature of 590 °C. A gas mixture consisting of hydrogen, evaporated trimethyl borate (TMBT, 1%), and carbon dioxide (0.2%) at a pressure of 30 Pa was used to maintain the growth of the diamond layer, using a gas phase with C_B/C_ = 312,500 ppm (resistivity = 0.017 Ω cm and n = 2.9 × 10^21^ cm^−3^ according to Hall measurements) [24].

Nanograss and Nanotip morphologies were formed on the surfaces of the BDD layers, using dry etching in radiofrequency (RF) plasma (PE-200, Plasma etch, Carson City, NV, USA) employing the self-masking nanostructuring process previously described in [19]. The Nanograss morphology was formed by etching for 60 min in O_2_ plasma (3 Pa, 200 W), and Nanotips were etched by a 30 min etching process in H_2_/N_2_ plasma (6 Pa, 500 W). All the samples were washed in a solution of NH_4_, H_2_O_2_, and demineralised water (1:1:4) for 10 min at 80 °C to remove amorphous carbon from the surfaces.

One-third of the samples were hydrogenated in a linear antenna MW plasma CVD system in pure hydrogen plasma for 60 min under the following conditions: a microwave power of 2 × 1.7 kW; a substrate temperature of about 330 °C, and a total gas pressure of 15 Pa. The second third of the samples were fluorinated in a capacitively coupled plasma reactive ion etching (CCP-RIE) Phantom System (Trion Technology, Clearwater, FL, USA) at a discharge power of 100 W, an exposure time of 30 s, a pressure of 20 Pa, and a CF_4_ flow of 20 cm^3^ s^−1^. The last third of the samples were oxidised in RF plasma for 1 min at 6 Pa and 200 W (PE-200, Plasma Etch, Carson City, NV, USA).

### 2.3. Substrate Preparation for MALDI-MS Analysis

The BDD/Al_2_O_3_ substrates were selectively passivated using a silicone-based screen-printing paste with mineral filler 240-SB (FERRO, Mayfield Heights, OH, USA) and a semi-automatic printing machine (TY-600H, ATMA, Taoyuan City, Taiwan) and were contacted with Ag paste (AST6025, Sun Chemical, Parsippany, NJ, USA) and Cu wiring to create a reusable target plate for MALDI-MS analyses. The whole BDD preparation procedure is schematically depicted in Figure 1.

### 2.4. Scanning Electron Microscopy

The surface morphologies of the fabricated diamond films/structures were analysed using a field-emission scanning electron microscope (FE-SEM) operating in secondary electron mode (MAIA3, Tescan Ltd., Brno, Czechia), using a beam energy of 10 keV. 

### 2.5. Raman Spectroscopy

The Raman spectra of the prepared samples were acquired using a Renishaw InVia Reflex Raman spectrometer (New Mills, UK) equipped with a He-Cd laser, with an excitation wavelength of 442 nm, and an air-cooled CCD detector. For the measurements, we used a Leica microscope objective with a 100× magnification (NA = 0.9), and the total laser power at the total exposure of the sample was 8.25 mW. The accumulation time was set at 30 s for all the measurements. Before the measurements, the Raman spectrometer was calibrated based on the crystalline diamond peak (1332 cm^−1^). 

### 2.6. Contact Angle Measurements

The wetting properties of the as-grown and surface-terminated (H-, O-, and F-) diamond film surfaces were determined by contact angle measurements at room temperature using a static method and a material–water droplet system. A water droplet of 3 μL in volume was dropped onto the investigated diamond surface, and an image of the formed drop was captured using a digital charged-coupled device camera. The contact angle, used to assess the wetting properties of the surface, was calculated using the Surface Energy Evaluation (SEE) System (Masaryk University, Brno, Czechia) for the multipoint fitting of the drop’s profile.

### 2.7. Photoelectron Spectroscopy 

X-ray photoelectron spectroscopy (XPS) spectra and ultraviolet photoelectron spectroscopy (UPS) spectra were obtained using an AXIS-Supra photoelectron spectrometer (Japan) equipped with monochromatised Al Kα radiation (1486.7 eV) and a He I light source (21.2 eV). The measurements were carried out at a photoelectron emission angle of 90°, an X-ray power of 150 W, an energy step of 0.1 eV, and a pass energy of 10 eV, ensuring an overall energy resolution of 0.45 eV for the high-resolution core-level spectra. To quantify the atomic compositions, integrated high-resolution core-level peak areas were analysed after Shirley’s electron inelastic background subtraction was performed. This analysis was performed with ESCApe software (ver. 1.4.0.1149, Kratos Analytical Ltd., Manchester, UK), which incorporates atomic sensitivity factors. The UPS spectra were measured with a 55 μm aperture and a bias of 9 V (the spectra were calibrated to compensate for the applied bias). 

### 2.8. Sample Preparation

The DHB matrix solution was prepared at 20 mg mL^−1^ in TA30 (30% ACN/70% H_2_O/0.1% TFA). Peptide stock solutions were prepared at 1 mg mL^−1^ in TA30 and then diluted to final concentrations of 100 nM, 10 nM, and 1 nM with TA30 for each peptide. The details about the peptides used in this study are provided in Table 1 and Appendix A. For the analysis of the peptides via MALDI-MS, 1 µL of the samples were spotted on Ground Steel *800/384 TF* MALDI targets (Bruker Daltonics, Billerica, MA, USA) or Continuous BDD, BDD Nanograss, and BDD Nanotip surfaces and mixed with 1 µL of the DHB matrix solution; the spots were dried at lab temperature.

The 3D structures of the peptides (Appendix A) were generated from PDB files, which were computed using the online tool I-TASSER (Iterative Threading ASSEmbly Refinement, version 5.1; University of Michigan, Ann Arbor, MI, USA) through the webpage of Yang Zhang’s research group [26,27,28,29]. The 3D images of the peptides were visualised using Deep View/Swiss–PdbViewer (version v4.1.0, Swiss Institute of Bioinformatics, University of Lausanne, Lausanne, Switzerland), showing the peptide backbone and the surface of the peptide using electrostatic potential values. The Brad and Ang II peptides are too short to generate pdb files.

### 2.9. MALDI-MS Analysis

The samples were analysed using an UltrafleXtreme II mass spectrometer (Bruker Daltonics, MA, USA). The spectra were acquired in the reflectron positive ion mode, collecting 6000 shots from 3 different spots for all the measurements. The laser parameters were as follows: a wavelength of 337 nm, a frequency of 1 kHz, and a laser power of 70%. A mass window of *m*/*z* 500–3500 was selected for the analysis of the peptides. The data were acquired with FlexControl 3.4 and processed with the FlexAnalysis 3.4 software program (both from Bruker Daltonics, Billerica, MA, USA) using the following processing parameters: The peak detection algorithm was Snap, the signal-to-noise threshold = 6, and the quality factor threshold was 30. 

## 3. Results and Discussion

### 3.1. SEM Analysis of BDD

Figure 1 shows representative SEM images of the as-grown (Continuous) and structured (Nanograss and Nanotip) BDD films. The Continuous film (the first row in Figure 1) reveals the typical nanocrystalline characteristic of a diamond film, with a grain size in the range from about 20 to 200 nm. The diamond film’s thickness is about 4.6 μm, with a quite smooth surface. After the structuring processes, Nanograss- and Nanotip-like surface features (the second and the third rows in Figure 1, respectively) were fabricated. In the case of the Nanograss structure, an approximately 300 nm thick top layer has been converted to a dense “forest” of sharp diamond peaks also cleaned from graphitic phases (see Figure 1, second row, nanograss, angular view = 45°). The overall thickness of the film remained the same. The Nanotip structure has not revealed such deep and sharp structures on the top as the Nanograss sample. Moreover, the total film thickness decreased to 3.5 μm because of different structuring process parameters. In general, the RIE-structuring process involves both physical etching by energetic ion collision and chemical etching by reactive atoms/ions, which influence the morphologies of the nanostructured surfaces. In both cases, the observed structured morphologies fit well with the published data. Anisotropic etching occurs for different crystallographic planes of polycrystalline diamond, depending on (i) the gas environment used (O_2_ vs. H_2_/N_2_) and (ii) the ratio of the diamond’s sp^3^ phases to sp^2^ carbon phases localised mainly at the grain boundaries [30]. However, it is important to note that after the RIE process, both as-etched nanostructures (Nanograss and Nanotips) need to be washed in a solution of NH_4_, H_2_O_2_, and demineralised water (1:1:4) to remove the formed and/or re-deposited amorphous carbon on the diamond film’s surface [19].

### 3.2. Raman Analysis of BDD

It is important to note that structuring processes have a negligible effect on the bulk chemical compositions of the BDD films, as confirmed by the Raman measurements in Figure 2. This observation is consistent with findings for highly boron-doped diamond with similar dimensions of surface structures [31]. The Raman spectra of all three kinds of samples reveal the same features, with the characteristic Fano-shaped highly boron-doped diamond bands around 460 cm^−1^ (B_1_) and 1200 cm^−1^ (B_2_) [32,33]. The zone-centre phonon mode of the cubic diamond (at 1290 cm^−1^ (ZCP_D_)) is red-shifted from its original position in intrinsic diamond (at 1332.9 cm^−1^) because of the ‘Fano’ effect and phonon confinement caused by incorporated boron atoms [34]. The band at around 1530 cm^−1^ (G-band) is attributed to the graphitic sp^2^-hybridised carbon phases. The presented Raman spectra are characteristic of boron-doped diamond exhibiting metallic electrical conductivity [34].

### 3.3. Contact Angle Measurements of BDD

The water contact angle (CA) measurements are shown in Figure 3 and illustratively present the effects of the diamond surface terminations under different conditions. The as-grown Continuous BDD diamond films reveal hydrophobic properties, with a contact angle of about 69°. After the structuring process (Nanotip fabrication), the CA decreased to about 10°. Further, the plasma-based oxidisation of the diamond surface revealed unambiguously hydrophilic properties, with a contact angle value lower than 4°. After the hydrogen plasma treatment, the diamond film exhibited strong hydrophobic properties (a contact angle of about 90°). Finally, after the fluorination process, the diamond film exhibited super-hydrophobic properties (a contact angle of more than 120°). Within the different sets of structures (Continuous, Nanograss, and Nanotips) some differences in the value of the CA were observed because of an additional effect of the diamond film’s surface roughness; however, the trends remain the same. 

### 3.4. XPS Measurements of BDD

The atomic compositions of the BDD sample surfaces (before peptide absorption) were derived using XPS measurements. The compositions are outlined in Table 2. The representative XPS survey spectra are shown in Figure 4. The presence of the p-type boron dopant in nanocrystalline diamond substrates, with concentrations ranging from 1 at.% to 3 at.% (10,000–30,000 ppm), was confirmed in both the as-grown Continuous and structured samples with different surface terminations. XPS identified various bonding types of boron, including B-C and B-O bonds, on the samples’ surfaces. Consequently, B atoms could be present as a dopant within the nanocrystalline diamond grains as well as an adsorbent on the diamond’s surface. The concentrations of O (8.3 at.%) and N (1.1 at.%) on the surface were relatively high in the as-grown Continuous samples and increased further after the structuring process, particularly for the Nanograss and Nanotip samples. The appearance of Si and Al contaminants is attributed to signals originating from the substrate beneath the nanocrystalline diamond layers.

Following the H and F plasma treatments, the concentrations of the O and N contaminants decreased. However, these treatments, along with the etching of the Continuous film and the surface plasma treatment, led to an increase in sample surface roughness and potential variation in the BDD film’s thickness on the sample surface. Consequently, the concentrations of Si and Al increased on the -H- and -F-terminated surfaces, respectively. It is noteworthy that the high concentration of O (21.9 at.%) on the surface of the Nanograss-H sample is associated with an increase in the Si concentration (13.0 at.%) on the surface.

In the O-terminated samples, high oxygen concentrations were measured on the sample surfaces (note, with lower concentrations of silicon and aluminium atoms from the substrate). Oxygen concentrations were higher for the structured Nanograss-O (20.4 at.%) and Nanotip-O (17.4 at.%) samples, with enhanced sample surface areas, compared to the Continuous BDD-O (9.0 at.%) sample. A similar trend was observed for the F-terminated samples, where the F concentrations were higher for the Nanograss-F (24 at.%) and Nanotip-F (27.8 at.%) samples than for the Continuous BDD-F sample (17.5 at.%).

### 3.5. UPS Measurements of BDD

Distinct surface terminations led to surface dipole charge variations on the BDD sample surfaces, as reflected by changes in surface zeta potentials or electron affinities [35]. Specifically, a negative electron affinity (NEA) or a positive electron affinity (PEA) was observed on H-terminated or on O- and F-terminated diamond surfaces, respectively. The UPS measurements of the BDD samples confirmed an NEA for only the H-terminated samples independent of the surface structuring (see Table 3 and Appendix A).

The valence band maxima (VBM) and the cutoff energies were determined by extrapolating the UPS spectral edges to the zero-intensity levels. The electron affinity (EA) was computed as follows:EA = WF + E_VBM_ − E_BG_(1)
where the work function (WF = 21.2 − E_cutoff_) was computed from the cutoff energy of the valence band spectra and excitation light’s energy, E_VBM_ is the extrapolated edge of the valence band maxima, and E_BG_ is the bandgap of the diamond. The measured and computed values are included in Table 3. The negative EA was derived for the H-terminated samples, whereas the positive EA was derived for the as-grown Continuous and -F- and -O-terminated samples. 

These findings are in good agreement with the expected impact of the formed surface dipole on the diamond’s band alignment. When the diamond surface is terminated with non-diamond elements, in our case hydrogen, oxygen, or fluorine, the charge dipole is formed because of the differences in the electronegativity values (δ) of these elements (carbon (2.55), hydrogen (2.2), oxygen (3.44), and fluorine (3.98)). In the case of the H-terminated diamond surface, different values of δ result in a positive potential drop (C^δ-^-H^δ+^) along the C-H bond, which pushes the surface vacuum level down, ultimately leading to a negative electron affinity, in our case, ranging from −0.3 to −0.5 eV. It should be noted that this value should be as high as −1.3 eV for C(111) single-crystal diamond [36]. For the O/F terminations, the opposite dipole (C^δ+^-X^δ-^) is formed, and the characteristic values of the EA are positive. The Continuous BDD/H samples also revealed positive values of EA, which should be attributed to the washing procedure used to remove the amorphous carbon from the surface.

### 3.6. MALDI-MS Analysis of Peptides 

For the initial MALDI-MS measurements, 0.5 µL of the peptide mixture (Peptide Calibration Standard II; Bruker, Billerica, MA, USA) (Table 1), at a concentration of 0.2 pmol μL^−1^, was spotted on the BDD surfaces (Figure 5) or Ground Steel MALDI chip’s surface. The sample was mixed with the matrix and dried at room temperature. Finally, the analysis was performed using MALDI-TOF. The MALDI-MS workflow, together with representative spectra for a mixture of all the peptides, is shown in Figure 6. It has to be noticed that strong peaks corresponding to sodium adducts are observed in the Ground Steel MS spectra, while the spectra for the BDD surfaces (Continuous and Nanograss) possess weak adduct signals, e.g., for Ground Steel, 61.5% were protonated adducts and 38.5% were sodium adducts; for Continuous BDD, 96.4% were protonated adducts and 3.6% were sodium adducts; and for Nanograss BDD, 96.3% were protonated adducts and 3.7% were sodium adducts. Interestingly, the BDD surfaces reduced the sodium adduct contamination significantly (Figure 6).

Further characteristics of the peptides (i.e., the percentages of the classes of amino acids present within the peptides) are shown in Appendix A. The graphical representation of the same information is shown in Figure 7. In Figure 7, there is also depicted the average percentages of the hydrophobic amino acids present in Brad, Sub P, and Ren S, as indicated by dashed lines (average values for Brad, Sub P, and Ren S peptides).

Then, individual peptides (Table 1 and Appendix A) from small to large sizes, with *m*/*z* from 700 to 3200, were measured using MALDI-MS to investigate the efficiency of their detection by MALDI-MS. Ang II is a peptide that plays a key role in balancing the renin–angiotensin system [37]; a dysregulation can cause cardiovascular diseases [38]. Ang I and Ren S both can be used as potential molecular biomarkers for the treatment and diagnosis of cardiovascular diseases [39,40]. The nanostructured BDD Nanotip surface terminated by -H reached a limit of detection (LOD) of 12.2 pM for Ang II, with an *m*/*z* value of 1047 (Table 4). Moreover, Ang I, with an *m*/*z* value of 1296, was detected on the BDD Nanotip surface terminated by -H, reaching an LOD of 16 pM (Table 4). Ren S, with an *m*/*z* value of 1759, reached an LOD of 3.1 pM using the BDD Nanograss-modified interface terminated by -H (Table 4). 

Interestingly, other studies using LC-ESI-MS for the detection of Sub *P* (LOD = 4.4 nM [41], Ang I (LOD = 3.0 nM) [42], and Ang II (LOD = 3.7 nM) [42] and using LC coupled with tandem MS for the detection of Bomb (LOD = 12.3 nM) [43], revealed much higher LODs compared to our study (Table 4). Furthermore, the advantages of MALDI-MS are the easy operation and high-throughput approach.

When considering other peptides, especially Sub P, with an *m*/*z* value of 1348, could be detected at very low LODs at several interfaces, e.g., 6.2 pM (Continuous BDD terminated by -H), 2.3 pM (Continuous BDD terminated by -O), and 8.1 pM (BDD Nanotips terminated by -H) (Table 4). Sub P is a neuropeptide that acts as a neuromodulator; its increased levels are correlated to Parkinson’s disease. Thus, it is considered as a promising biomarker for Parkinson’s disease [44]. The LODs for all the combinations of peptides and BDD interfaces, together with Ground Steel, are summarised in Table 4. This study demonstrated small background signals, high sensitivities, and very low LOD values. In recent studies reported in the literature, Cournut et al. [1] used MALDI-MS and SALDI-MS to investigate the desorption/ionisation efficiencies of peptides on perfluorosilane-nanostructured steel plates. However, that study did not describe the sensitivities or LOD values of the peptides [1]. 

To quantify the beneficial effects of BDD-modified interfaces for the analysis of peptides, for every combination of peptide–BDD interfaces, the LOD_ratio_ was calculated as LOD_ratio_ = LOD_steel_/LOD_BDD_. The higher the LOD_ratio_ value, the better the performance of the peptide analysis using MALDI-MS on BDD interfaces in comparison to commercially available Ground Steel chips. In Table 4, we can identify especially three peptides, which showed significantly enhanced LOD_ratio_ values exceeding the value of 5, e.g., for the Sub P peptide on the Continuous BDD with -H termination (LOD_ratio_ = 7.03), on the Continuous BDD with -O termination (LOD_ratio_ = 19.19), and on the BDD Nanotip surface with -H termination (LOD_ratio_ = 5.42) (Table 4). Among all the other peptide–BDD combinations, only two peptides could be detected at a significant LOD_ratio_, e.g., Brad on Continuous BDD terminated by F- (LOD_ratio_ = 5.92) and Ren S on BDD Nanograss terminated by H- (LOD_ratio_ = 14.94) (Table 4).

To better understand why only those three peptides could be detected at such high LOD_ratio_ values and on which surfaces, additional analyses were performed. Three-dimensional (3D) plots showing the dependence of the LOD_ratio_ on the *m*/*z* and charge of the peptides were prepared (Figure 8). From these graphs, it is clear that enhanced LOD_ratio_ values of >5.0 were observed for Continuous BDD films terminated by all three elements (-H, -O, and -F; upper row in Figure 8), for BDD Nanograss terminated by -H (the middle image in the middle row in Figure 8), and for BDD Nanotips terminated by -H (the middle image in the lower row in Figure 8). The correlation of the BDD samples revealed enhanced LOD_ratio_ values with the electron affinities of their surfaces. The electron affinities of all the BDD-modified surfaces vary from +1.8 to −0.7 eV (Table 3). A closer inspection confirms that the BDD surface terminations, which revealed enhanced LOD_ratio_ values, exhibit moderate electron affinity values, e.g., −0.7 eV (Continuous BDD, -H), −0.5 eV (BDD Nanotips, -H), −0.3 eV (BDD Nanograss, -H), +0.8 eV (Continuous BDD, -O), and +1.0 eV (Continuous BDD, -F). If the electron affinity exceeded the value of +1.4 eV, all the LOD_ratio_ values were lower than 5.0, indicating that there might be an upper limit of the electron affinity (i.e., ~1.0÷1.4 eV), beyond which, we could not observe an LOD_ratio_ value of >5.0 for any peptide used in this study. This might indicate that moderate electron affinities of BDD samples, i.e., electronegativity dipole drops at surfaces, are beneficial for effective peptide analysis using MALDI-MS. 

It is very important to also understand which features peptides need to have to be detected on BDD surfaces at enhanced LOD_ratio_ values. Appendix A indicates that the three peptides, Brad, Sub P, and Ren S, effectively detected at BDD interfaces using MALDI-MS, exhibit high percentages of hydrophobic amino acids (54.5–57.2%) in combination with low percentages of acidic amino acids (0–7.1%). Additional calculations showed that these three amino acids, Brad, Sub P, and Ren S, exhibited very low hydrophobicity values (10.03–14.50 kcal mol^−1^). At the same time, these three amino acids, Brad, Sub P, and Ren S, exhibited low charge densities at pH 7.4 (from −0.20 to +1.18) (Table 1). Appendix A also shows the distributions of the negative charges, positive charges, and hydrophobic patches on the surfaces of the peptides. So, we can assume that especially amino acids with high hydrophobicities and low charge densities are effectively detected using MALDI-MS. 

We can conclude that the reason especially Brad, Sub P, and Ren S were effectively detected on the surfaces of some BDD samples is the hydrophobic interaction between the BDD surface (represented by electron affinity values, e.g., from −0.3 eV to +1.0 eV) and hydrophobic peptides. 

It is also important to understand the optimal surface termination and morphology of the BDD for the MALDI-MS analysis of highly hydrophilic (highly acidic or highly basic) peptides. The results demonstrate that for the highly acidic AC18 peptide (e.g., negatively charged, with a charge, at pH 7.4, of −3.82 and a pI of 4.3) (Table 1 and Appendix A), the highest LOD_ratio_ value of 4.21 was obtained on the surface of the hydrophilic Continuous BDD terminated with -O (with an electron affinity of +0.8 eV and a negative electronegativity potential drop). The highly basic AC1 peptide (e.g., positively charged, with a charge, at pH 7.4, of +3.26 and a pI of 10.3) (Table 1 and Appendix A), the highest LOD_ratio_ value of 2.78 was obtained on the surface of the hydrophobic Continuous BDD terminated with -H (with an electron affinity of −0.7 eV and positive electronegativity potential drop). Thus, we can conclude that highly acidic and basic peptides can be detected on BDD surfaces having the opposite wetting properties, i.e., interfacial charge and electron affinity. 

The analysis of the peptides on the BDD surfaces could be affected also by other parameters of interfaces, such as the work functions (WFs) (Table 3) of the BDD interfaces. The dependence between the LOD_ratio_ value and WF for the Sub P peptide shows a linear decrease in the LOD_ratio_ value with increasing WF, with one exception (shown in the red circle in Figure 9). 

Thus, it seems that the values of the EA for BDD interfaces might be an important factor for the effective adsorption of peptides at BDD interfaces via hydrophobic interactions (Ren S, Brad, and Sub P) or via electrostatic interactions (AC1 and AC18). There are, of course, other types of interactions, which might be important for the adsorption of peptides at BDD interfaces, e.g., hydrogen bond interactions. The other factor that can contribute towards the effectiveness of the peptide analysis might be the WF of the BDD interfaces. It was already postulated that the main role of the matrix applied for MALDI-MS assays is to affect the work function of the metallic interface, resulting in a change in the electronic structure of the organic–metal interface and affecting both the kinetic energy and the flux of the released electrons [45,46]. The results shown here are only preliminary, revealing the key factors behind the effective peptide analysis on BDD surfaces using MALDI-TOF MS. Although some interfacial properties of the BDD are beneficial for the analysis of peptides, the main limitation of this study is the fact that we have not been able to finely tune the work function and electron affinity of a particular BDD surface. Thus, further investigation is still needed to better understand the complex interplay of the interfacial factors of the BDD and physicochemical properties of the peptides behind the peptide analysis on the BDD surface using MALDI-TOF MS. 

It is important to highlight that some of the peptides could be detected using MALDI-TOF MS on the BDD surface in a very sensitive fashion, with LODs as low as 2.3 pM, which is a prerequisite for the diagnosis of several diseases at early stages. In this preliminary report, our aim was not to apply such interfaces for the analysis of real samples but to, rather, understand the key factors behind the detection of peptides using mass spectrometry on boron-doped diamond surfaces. To prove practical utility of such interfaces for the analysis of real samples, further studies are needed. 

## 4. Conclusions

Our study has demonstrated, for the first time, the effective application of BDD chips for the detection of nine representative peptides using MALDI-MS. The two critical parameters for the effective detection of peptides on nanostructured BDD surfaces were identified: the type of atomic termination of the BDD surface and the hydrophobicity of the peptides. We found that BDD surfaces allowed for us to effectively detect hydrophobic amino acids at moderately processed BDD-nanostructured interfaces. At the same time, we observed that positively charged peptides could be detected on hydrophobic BDD surfaces and negatively charged peptides on hydrophilic BDD surfaces. According to these findings, we can conclude that to achieve the effective detection of peptides, it is necessary to tailor the BDD interface with a high affinity towards the target peptides. This means utilising hydrophobic interactions for hydrophobic peptides and electrostatic interactions for charged peptides. In future research, we intend to explore whether other biomolecules besides peptides can be effectively detected on BDD surfaces by MALDI-MS. Furthermore, we aim to determine the optimal interfacial properties (top surface morphology, conductivity, wetting properties, electron affinity, etc.) of BDD surfaces for such assays. Overall, our findings highlight the potential of BDD chips as a tailorable material platform for peptide detection and pave the way for further advancements in this field. Further work is, however, necessary to understand the complex interplay of the interfacial BDD interfaces in peptide adsorption and in the ability of such interfaces to release electrons for the ionisation of analyte molecules. Furthermore, BDD surfaces reduced sodium adduct contamination significantly.

## Data Availability

https://doi.org/10.5281/zenodo.12770402 (accessed on 6 January 2024).

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
