# Peer review of "What Are the Key Factors for the Detection of Peptides Using Mass Spectrometry on Boron-Doped Diamond Surfaces?"

_nanomaterials, 2024, doi:10.3390/nano14151241_

Round 1

Reviewer 1 Report

Comments and Suggestions for Authors

The manuscript “What are key factors for detection of peptides using mass spectrometry on boron-doped diamond surfaces?” (original article) deals with the investigation of the use of boron-doped diamond with different surface morphologies for detection of a small group of peptides by MALDI. The selected topic is actual and very interesting. Effective, sensitive and accurate analysis of peptides is a hot topic in the field of system biology, biochemistry, or analytical chemistry. The topic of the manuscript fits with the journal (Nanomaterials). The manuscript is appropriate for publication in the Journal Nanomaterials with minor revisions according to the comments listed below.

Comments:

Introduction:

*The information regarding the selection of the investigated peptides is missing.

*Benefits of MALDI in the analysis of peptides/proteins should be mentioned. The differences between ESI and MALDI should be discussed.

Section 2.8. Sample preparation:

*On-line tools I-TASSER and Deep View/Swiss-PdbViewer software are mentioned. The information regarding their version used are missing. Moreover, the manufacturer of the mentioned software is not mentioned (company, city, state).

Section 3.5. UPS measurements of BDD

*MALDI-MS analysis of peptides – the obtained LOD values should be compared with those ones investigated by ESI ionization technique by other working groups

*Please, modify the following sentence: “Let's correlate BDD samples revealing enhanced LODratio with electron affinity of their surfaces.” The scientific soundness of the sentence is very low.

Sections Discussion, Conclusions

*Limitations of the selected approach should be mentioned. Further strategies should be also presented. Analysis of real biological samples should be mentioned and critically evaluated.

Comments on the Quality of English Language

Minor editing of English language required.

Author Response

Please find the response in the attached file.

Reviewer 2 Report

Comments and Suggestions for Authors

The Authors studied the use of boron-doped diamond (BDD) based surfaces as sample plate material for the MALDI-MS measurement of peptides. They used 9 peptides of biological importance to test the suitability of the new surfaces. The results show that lower level of detection can be achieved for selected cases, but no universal improvement could be had. The work is novel, as not many publications use similar surfaces and the characterization of the new surface is also very thorough. 

There are only a few minor questions that are not clear from the manuscript. 

1. Sample preparation (chapter 2.8) states that you used 1 μL of sample but later you state that 0.5 μL were used (page 9, last paragraph). Please indicate why is this

2. Still to the topic of sample preparation, the Authors used mixed drop mode of sample prep. Since you change the topology of the surface, different spotting technique (sandwich for ex.) might give different results. Why was this not tried, although in some cases the target peptide was not detected?

3. Why only steel based measurements show Na adducts? Was the sample washing step different for the other samples and if so please include the differences in the manuscript.

4. BDD should allow for matrix-free measurement of peptides, in which case the interaction of the peptide with the surface, and the properties studied in this work should be more apparent. Why was this not tried? What would be the limit of detection in that case? This is also of interest if the BDD surface is ever used for MS-imaging, which could be important clinically.  

Author Response

(The authors gave the same response as above.)

Reviewer 3 Report

Comments and Suggestions for Authors

The manuscript by Aguedo et al. explores the use of boron-doped diamond (BDD) for the improved detection of peptides using MALDI-MS. While the manuscript is well-planned and written, there are some revisions necessary from this reviewer's perspective:

  1. The abstract needs to better contextualize the background and novelty of this study. These aspects are crucial for the reader to understand the quality of the study and decide if it merits further reading based on their aims.

  2. Certain parts of the methods section, particularly those describing the MALDI analyses, lack sufficient detail and should be expanded. Additionally, an expanded version or step-by-step protocol of the BDD steps would enhance the quality of this paper if provided as a supplementary file.

  3. Please rephrase and rewrite this part of the results to improve clarity:

"MALDI-MS analysis of peptides: For MALDI-MS measurements, 0.5 μL of the peptide mixture (Table 1) with a concentration of 0.2 pmol/μL was spotted on the BDD surfaces (Figure 5) or ground steel MALDI chip surface, mixed with matrix, dried at room temperature, and measured by MALDI-TOF. The MALDI-MS workflow, together with representative spectra for a mixture of all peptides, is shown in Figure 6. Further characteristics of peptides (e.g., percentage of amino acid classes present within peptides) are shown in Table S1, with a graphical representation in Figure 7. Figure 7 also depicts the average percentage of hydrophobic amino acids present in Brad, Sub P, and Ren S as a dashed line (average value for Brad, Sub P, and Ren S peptides)."

  1. It is not clear how the method tested represents an improvement over other current standard detection methods. This is a crucial aspect of this manuscript that the authors should emphasize. Please provide clear data on the detection of these peptides with an appropriate control and perform statistical analyses to demonstrate the extent to which the incorporation of BDD under different conditions improves the detection of the selected peptides.

  2. Some aspects of the discussion should be better focused. Additionally, the discussion section cannot be omitted; it should be mentioned alongside the results. This section needs to contextualize how the Raman and EM results compare to those obtained by other authors, as this comparison is currently missing in the manuscript.

Author Response

(The authors gave the same response as above.)

Round 2

Reviewer 3 Report

Comments and Suggestions for Authors

The authors have significantly improved the manuscript following the requested revisons and it can be now accepted.